# Do We Really Achieve Fairness with Explicit Sensitive Attributes?

## Abstract

Recent research on fairness has shown that merely removing sensitive attributes from model inputs is not enough to achieve demographic parity, as non-sensitive attributes can still reveal *sensitive information* [1] to varying extents. For instance, a person's "race" can be deduced from their "zipcode" to some extent. While current methods directly utilize explicit sensitive attributes (e.g., "race") to debias model predictions (e.g., obtained by "zipcode"), they often fail to uphold demographic parity. This is especially true for high-sensitive samples, whose non-sensitive attributes are more likely to leak sensitive information than low-sensitive samples. This challenge stems from the model treating each sample with a specific sensitive attribute, while the prediction only incorporates partial sensitive information, leading to potential biases. This observation highlights the need for demographic parity measurements that account for the degree of sensitive information leakage in individual samples, and differentiate between samples with varying degrees of leakage. To address this issue, we introduce a new definition of group fairness measurement called $\alpha$-Demographic Parity, which ensures demographic parity for samples with differing degrees of sensitive information leakage. To achieve $\alpha$-Demographic Parity, we propose to directly promote the independence of model predictions from the distribution of sensitive information, rather than the specific sensitive attributes. This approach directly minimizes the Hilbert-Schmidt Independence Criterion between the two distributions, thereby ensuring more precise and fair predictions across all subgroups. Our proposed method outperforms existing approaches in achieving $\alpha$-Demographic Parity and demonstrates strong performance in scenarios with limited sensitive attribute information, as evidenced by extensive experiments. Our code is anonymously available at https://anonymous.4open.science/r/TMLR_STFS_code-2ED6.

## 1 Introduction

Deep neural networks (DNNs) have found increasing use in high-stake decision makings processes such as credit scoring (Petrasic et al., 2017; Avery et al., 2012), criminal justice (Berk et al., 2021; Grgic-Hlaca et al., 2018), and healthcare (Rajkomar et al., 2018; Ahmad et al., 2020). However, recent literature (Mehrabi et al., 2019; Corbett-Davies & Goel, 2018; Du et al., 2021; Chuang & Mroueh, 2020) has highlighted that DNNs exhibit biases that can have significant implications for the fairness of the decisions. Given the growing concerns, there is much attention on accurately evaluating the extent of these biases.

Previous studies (Du et al., 2020; Dwork et al., 2012; Corbett-Davies & Goel, 2018) have shown that *sensitive information* of a data sample can be revealed explicitly by sensitive attributes or partially by non-sensitive attributes. For example, sensitive information about a person's race may be disclosed explicitly through an attribute labeled "race", or partially through an attribute like "zipcode". Therefore, simply removing the sensitive attributes from the model input may not guarantee to achieve demographic parity, as non-sensitive attributes can still, to varying extents, reveal the sensitive information (Kamishima et al., 2012).

---

[1] In this work, we use the term *sensitive information* to refer to *the degree to which we can infer demographic information from non-sensitive attributes*, and we use the term *sensitive attribute* to refer to the explicit sensitive attributes such as "race", "gender".

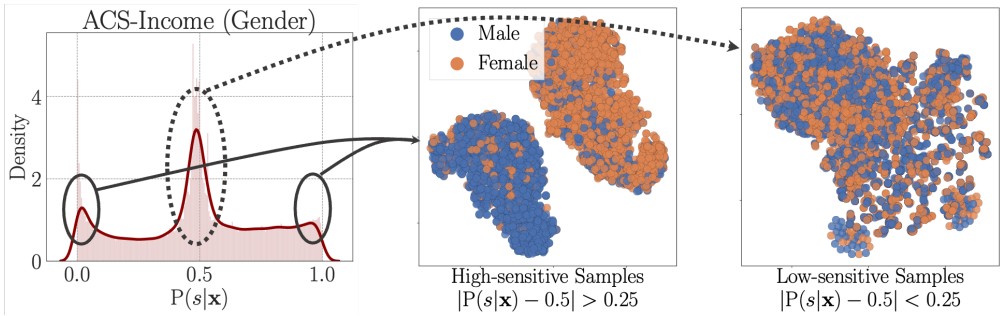

Figure 1: The distribution of sensitive information in non-sensitive attributes $P(s|\mathbf{x})$ (**Left**), and the two-dimensional representation of high-sensitive (◯, **Middle**) samples and low-sensitive (⬚, **Right**) samples. The high-sensitive samples are split into two distinct clusters based on gender, implying a greater level of sensitive information leakage. High-sensitive samples correspond to those samples at the two ends of the distribution, i.e., $P(s|\mathbf{x})$ is close to 0.0 or 1.0. Conversely, low-sensitive samples, characterized by $P(s|\mathbf{x})$ values close to 0.5, exhibit a mixture of genders, suggesting less sensitive information leakage.

Current fairness models aim to mitigate the impact of sensitive information in non-sensitive attributes ($\mathbf{x}$) by incorporating explicit sensitive attributes ($s$) to debias the model's predictions. However, these approaches implicitly assume that all non-sensitive attributes can leak sensitive information *to the same extent* across all samples, which may not always be the case. We consider the following example.

*Example 1.* Suppose we use a machine learning model to predict a candidate's acceptance in a college admissions scenario where the "zipcode" is used as a non-sensitive attribute. Since the "zipcode" leak different amounts of race information, the model trained on the biased data may treat different candidate groups with different race unfairly. Some zipcodes can leak race information, while some are not, due to residential segregation and population distribution patterns. Suppose we have two samples, one with *zipcode 1*, which entirely leaks the sensitive attribute, while another one with *zipcode 2* leaks no race information. Current methods treat both samples as if their non-sensitive attributes contain explicit race attributes. However, the non-sensitive attribute of the second sample does not reveal any race information. Employing explicit race attributes for debiasing would implicitly assume that all non-sensitive attributes across all samples leak sensitive information uniformly. This assumption may not hold true and emphasizes the importance of recognizing the varying degrees of sensitive information leakage present in non-sensitive attributes when seeking to debias machine learning models.

Yet, the presence of varying degrees of sensitive information in non-sensitive attributes calls for a more nuanced approach when attempting to debias machine learning models as follows:

> ***When non-sensitive attributes exhibit varying degrees of sensitive information leakage, how can we effectively account for this to ensure fairness in machine learning models?***

To investigate the extent to which non-sensitive attributes contain sensitive information, we conducted a preliminary study and presented the findings in Figure 1. Specifically, we trained a model to estimate the correlation between the sensitive attribute $s$ (i.e., gender) and non-sensitive attributes $\mathbf{x}$, represented by $P(s|\mathbf{x})$. We then plotted the density distribution of $P(s|\mathbf{x})$ across different samples, as shown in the left subfigure, where the x-axis indicates the value of $P(s|\mathbf{x})$ and the y-axis represents the normalized count of samples. The density distribution reveals that $P(s|\mathbf{x})$ varies across different samples, with high-sensitive samples (◯, middle figure) having values close to either 0.0 or 1.0, and low-sensitive samples (⬚, right figure) having values close to 0.5. This study clearly indicates that various samples' non-sensitive attributes contain different amounts of sensitive information.

In this study, we propose a new group fairness metric called $\alpha$-Demographic Parity to measure demographic parity at varying levels of sensitive information leakage. This metric ensures that each subgroup, defined by the level of sensitive information leakage, satisfies demographic parity. By achieving demographic parity at a finer granularity, $\alpha$-Demographic Parity ensures that all subgroups satisfy demographic parity, indicating that the demographic parity has been achieved across all values of $\alpha$. The proposed metric enables a more nuanced evaluation of demographic parity, allowing us to identify and address disparities in subgroups with

different levels of sensitive information leakage. $\alpha$-Demographic Parity can help improve the fairness of machine learning models and promote equity in decision-making processes.

To achieve $\alpha$-demographic parity, we propose to directly encourage the independence of the distribution of sensitive information and model predictions, which is intended to ensure that the model produces unbiased predictions regardless of the level of sensitive information present in the non-sensitive attributes. Specifically, we formulate this problem as a cross-task knowledge distillation task, where a sensitive teacher learns the distribution of the sensitive information, and a fair student learns the distribution of the prediction. We then enforce the independence between the teacher and the student by minimizing the Hilbert-Schmidt Independence Criterion between the two. In addition, our model can naturally tackle the limited sensitive attribution scenario since the teacher can be trained using partial samples with sensitive attributes. We highlight our **main contributions** as follows:

- Our study reveals that the levels of sensitive information present in non-sensitive attributes vary across all samples, which enables us to identify high-sensitive (most likely to leak sensitive information) and low-sensitive samples. We then demonstrate that demographic parity is more likely to be violated for high-sensitive samples than for low-sensitive ones, indicating that current fairness approaches may not be able to guarantee demographic parity for high-sensitive samples.

- We propose a novel approach, STFS, which leverages the distribution of the sensitive attributes to constrain the prediction for fairness via a cross-task [2] knowledge distillation framework. The sensitive teacher is designed to extract sensitive information from non-sensitive attributes, while the fair student makes fair predictions for downstream tasks. The independence between the distribution of the prediction and the sensitive information is guaranteed by minimizing the Hilbert-Schmidt Independence Criterion between the two.

- We conduct experiments on various datasets to validate the effectiveness of the proposed STFS. The experimental results show that our proposed method achieves the most favorable accuracy-fairness trade-off for both high-sensitive samples and low-sensitive samples. For the scenario of limited sensitive attributes, the experimental results show that our method can achieve comparable fairness performance with less than 20% training samples.

It is important to highlight that our proposed method differs from existing fairness approaches tailored for datasets with limited sensitive attributes (Abraham et al., 2019; Zhao et al., 2021; Dai & Wang, 2021; Du et al., 2021). Our method distinctively concentrates on addressing varying degrees of sensitive information present in non-sensitive attributes by employing a sensitive teacher to determine the level of sensitive information. This strategy allows the fair student model to generate fair predictions for downstream tasks. Although our method can be applied to datasets with limited sensitive attributes, this advantage is supplementary to our primary objective. The main goal of our approach is to tackle the challenges arising from varying levels of sensitive information in non-sensitive attributes, enabling more fair predictions across all subgroups.

## 2 Motivation

In this section, we present preliminary experiments to motivate our work. Specifically, we investigate the distribution $P(s|\mathbf{x})$, which shows the leakage of sensitive information from non-sensitive attributes varies across different training samples. In particular, we discover that the violation of demographic parity primarily occurs in high-sensitive samples. Therefore, we introduce a new concept called $\alpha$-Demographic Parity to measure group fairness with a $\alpha$-specific level of sensitive information leakage.

**Notations.** In this paper, we consider a dataset represented as $\{(\mathbf{x}i, s_i, y_i)_{i=1}^N\}$, where $N$ denotes the total number of samples. For each sample $i$, the non-sensitive attributes are represented by $\mathbf{x}_i \in \mathbb{R}^d$. The binary sensitive attribute is denoted by $s_i \in \{0, 1\}$, and the label of the downstream task is represented by $y_i \in \{0, 1\}$. We use $\hat{y} \in [0, 1]$ to signify the predicted probability of the downstream task, which is obtained from a machine learning model $f(\mathbf{x}) : \mathbb{R}^d \to [0, 1]$ with trainable parameter $\theta$. For clarity and conciseness, we exclusively consider binary classification and binary sensitive attributes in this paper.

---

[2]By *cross-task*, we mean the teacher and student learn different tasks.

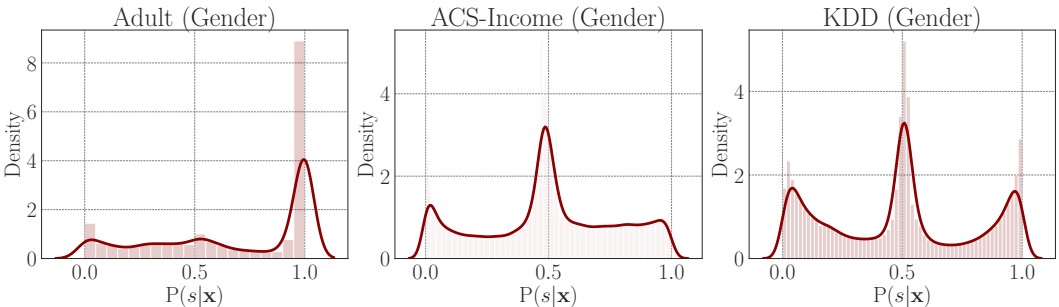

Figure 2: The distribution of $P(s|\mathbf{x})$. The $P(s|\mathbf{x})$ value reflects the amount of *sensitive information* ($s$ refers to gender in this example) contained in $\mathbf{x}$. The distribution shows that sensitive information in non-sensitive attributes varies over different samples. More results are presented in Appendix B.1.

**Demographic Parity (DP).** DP requires the predictions $\hat{y}$ to be independent of the sensitive attribute $s$, i.e., $P(\hat{y}|s=0) = P(\hat{y}|s=1)$. The existing approach to achieve algorithmic fairness is to ensure independence between the sensitive attribute $s$ and the prediction $\hat{y}$, by leveraging the explicit sensitive attributes (i.e., $s$) to debias the machine learning model with non-sensitive attributes (i.e., $\mathbf{x}$) as input. Given the difficulty of optimizing the independence constraints, previous works (Madras et al., 2018; Agarwal et al., 2018; Wei et al., 2019; Taskesen et al., 2020) propose the relaxed regularization $\Delta\mathrm{DP}(f) = |\mathbb{E}_{\mathbf{x}\sim P_0}f(\mathbf{x}) - \mathbb{E}_{\mathbf{x}\sim P_1}f(\mathbf{x})|$ to penalize the cross-entropy loss for the downstream task, where $\mathbb{E}_{\mathbf{x}\sim P_{0/1}}f(\mathbf{x})$ is approximated by the average of the predictions for samples with sensitive attribute 0/1.

### 2.1 Problems of the Current Practice for Fairness

Existing fairness approaches employ the explicit sensitive attribute $s$, largely depending on the assumption that non-sensitive attributes can disclose information about the sensitive attribute. Nonetheless, our analysis reveals that the presence of sensitive information in non-sensitive attributes varies across distinct samples. To substantiate our argument, we examine the quantity of sensitive information embedded in non-sensitive attributes. In particular, we develop a model designed to probe sensitive information within non-sensitive attributes, which accepts the non-sensitive attribute as input and predicts $P(s|\mathbf{x})$. The distribution of $P(s|\mathbf{x})$ is illustrated in Figure 2. **Finding 1: The presence of sensitive information in non-sensitive attributes differs among various samples.**

Further, we also conduct experiments to investigate the violation of demographic parity for data samples with different values of $P(s|\mathbf{x})$ and present the results in Figure 3. We first split the data samples into two groups: high-sensitive samples ($|P(s|\mathbf{x}) - 0.5| > 0.25$) and low-sensitive samples ($|P(s|\mathbf{x}) - 0.5| < 0.25$). We plot the distribution of the prediction probability of different demographic groups. The results in Figure 3 show that both the unfair and fair models exhibit greater violation of demographic parity for high-sensitive samples compared to low-sensitive samples. Specifically, the unfair model has a $\Delta\mathrm{DP}$ of 0.187 for high-sensitive samples and 0.008 for low-sensitive samples, while the fair model has a $\Delta\mathrm{DP}$ of 0.064 for high-sensitive samples and 0.003 for low-sensitive samples. Thus we had **Finding 2: The violation of demographic parity primarily occurs in high-sensitive samples.**

Moreover, from the results in Figure 3(b), in the fair model [3], the violation of demographic parity is more severe for high-sensitive samples, with a violation of 0.064 compared to 0.003 for low-sensitive samples, indicating a significant difference in demographic parity violation. **Finding 3: the current fair model cannot achieve demographic parity for high-sensitive samples.**

The violation of demographic parity occurs because the current approach does not consider the level of sensitive information in non-sensitive attributes. To address this issue, we propose a new definition of group fairness and the associated metric in the next section.

---

[3]The fair model in this experiment uses fairness constraints for fairness learning

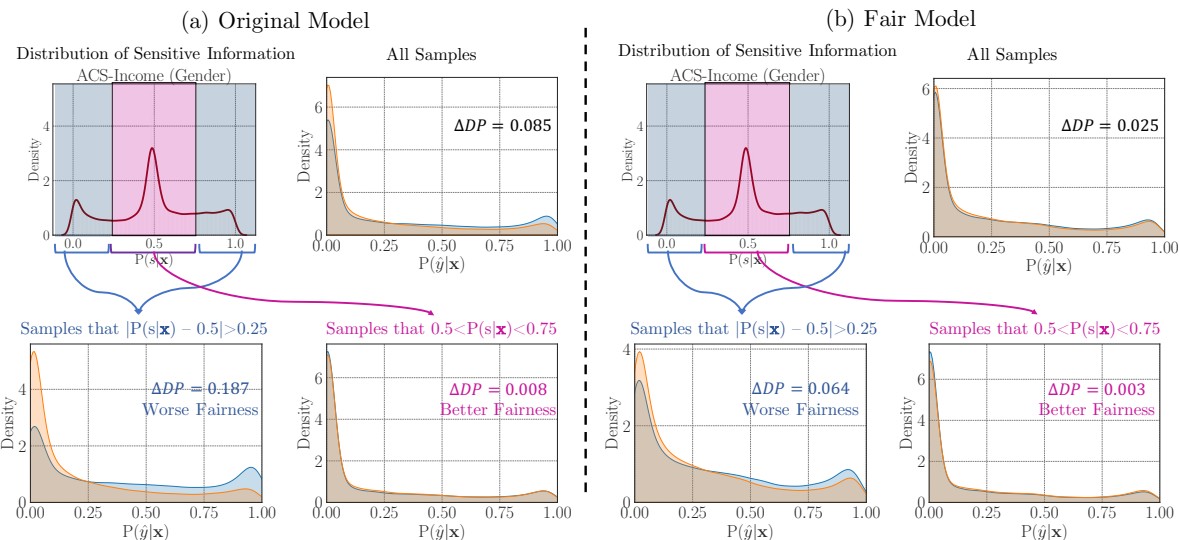

Figure 3: The $\Delta$DP on high-sensitive samples and low-sensitive samples. For the original (unfair) model shown in subfigure (a), the results indicate that $\Delta$DP of high-sensitive samples is significantly larger than that of low-sensitive samples. Interestingly, for low-sensitive samples, there is no fairness issue as the $\Delta$DP is close to 0. For the fair model shown in subfigure (b), the results show that even when the overall $\Delta$DP is relatively low (0.025), the high-sensitive samples still experience fairness issues as their $\Delta$DP is quite high (0.064). The sensitive attribute is "gender".

## 2.2 $\alpha$-Demographic Parity

Based on our preliminary experimental findings, we have gained two key insights: *(i)* demographic parity violations primarily occur in high-sensitive samples, and *(ii)* current fairness methods are not effective in achieving demographic parity for high-sensitive samples. These findings suggest that current demographic parity metrics are inadequate for measuring fairness, as they do not account for the varying levels of sensitive information in non-sensitive attributes.

To address this limitation, we propose the following definitions to measure fairness for different levels of sensitive information. We take into account the amount of sensitive information leaked by a sample and treat samples with varying degrees of information leakage differently, achieving a more nuanced understanding of fairness. First, we define a group that specifies the level of sensitive information leakage as follows:

**Definition 2.1** ($\alpha$-Sensitive Group)**.** *An individual sample belongs to the $\alpha$-sensitive group if $P(s|\mathbf{x}) \in [0, \alpha] \cup [1 - \alpha, 1]$, where $\alpha \in (0, 0.5]$*

where $\alpha$ denotes the level of difficulty in inferring sensitive attributes from non-sensitive attributes. A lower $\alpha$ value means the sample contains more sensitive information. For example, if $\alpha = 0.1$, the samples in $\alpha$-Sensitive Group satisfy $P(s|x) \in [0, 0.1] \cup [0.9, 1]$. The samples are more likely to reveal more sensitive information since it is easy for them to infer sensitive information from their non-sensitive attributes.

**Definition 2.2** ($\alpha$-Demographic Parity)**.** *A machine learning model satisfies $\alpha$-Demographic Parity if $\forall \alpha \in [0, 0.5]$, $\alpha$-Sensitive Group satisfy $P(\hat{y}, s|\mathbf{x}) = P(\hat{y}|\mathbf{x})P(s|\mathbf{x})$.*

Our proposed definition of group fairness separates individuals into different groups based on their level of sensitive information leakage and ensures demographic parity within each group. It is worth noting that when $\alpha = 0.5$, all samples are in the same group, and $\alpha$-Demographic Parity degrades to demographic parity. To measure $\alpha$-Demographic Parity, we propose the $\alpha$-$\Delta$DP metric to evaluate the violation of $\alpha$-Demographic Parity as follows:

$$\alpha\text{-}\Delta DP = \left| \frac{\sum_{i=1}^{N_0} P(\hat{y}_i|s_i=0)}{N_0} - \frac{\sum_{i=1}^{N_1} P(\hat{y}_i|s_i=1)}{N_1} \right|, \quad \text{if} \quad P(s_i|x_i) \in [0, \alpha] \cup [1-\alpha, 1], \tag{1}$$

where $N_0/N_1$ is the number of samples with the sensitive attribute equal to 0/1 in the $\alpha$-Sensitive Group.

In addition, we define the expectation of $\alpha$-$\Delta DP$ as $\mathbb{E}_\alpha\,\alpha$-$\Delta DP$ over $\alpha \in (0, 0.5]$ to measure the violation of $\alpha$-Demographic Parity. A lower $\alpha$-$\Delta DP$ indicates lower violation of $\alpha$-Demographic Parity, and vise versa. In our experiment, we use $\mathbb{E}_\alpha\,\alpha$-$\Delta DP$ as our fairness metric. Specifically, we compute the average of $\alpha$-$\Delta DP$ over a number of sampled $\alpha$ values to approximate $\mathbb{E}_\alpha\,\alpha$-$\Delta DP$.

## 3 Methodology

In this section, we present our proposed method (STFS), formulated as a cross-task distillation model. We provide a theoretical analysis to demonstrate the guarantee of $\alpha$-Demographic Parity. We discuss the benefits of STFS in scenarios with limited sensitive attributes and their relationship to demographic parity.

### 3.1 The Proposed Method - STFS

The preliminary experimental results indicate that previous fairness methods cannot achieve demographic parity for high-sensitive samples. This is because sensitive information varies among samples, but existing fair methods use explicit sensitive attributes. These observations motivate us to explore further solutions. We propose to directly promote the independence of the distribution of sensitive information (instead of sensitive attributes) and predictions. The main idea of the STFS is to enforce independence between the distribution of sensitive information and the prediction of the downstream task.

**Overview of STFS.** We formalize the problem as cross-task knowledge distillation, which involves a **S**ensitive **T**eacher model and a **F**air **S**tudent model (STFS). Since the distribution of sensitive information is not explicitly available in the dataset, we use a teacher model to predict the sensitive information, i.e., the sensitive teacher. The sensitive teacher learns the distribution of sensitive information in non-sensitive attributes, while the fair student learns the distribution of downstream task predictions. We encourage the independence between the outcomes of the sensitive teacher and fair student by enforcing the Hilbert-Schmidt Independence Criterion to be zero. The proposed method, STFS, is illustrated in Figure 4. Our goal is to achieve $\alpha$-Demographic Parity, which requires $P(\hat{y}, s|\mathbf{x}) = P(\hat{y}|\mathbf{x})P(s|\mathbf{x})$. To achieve this goal, we utilize the sensitive teacher model (the red model in Figure 4) to model the distribution $P(s|\mathbf{x})$ of sensitive information in the non-sensitive attributes, which is $\hat{s} = f_t(\mathbf{x}, \theta_t) = P(s|\mathbf{x})$, where the trainable parameters are $\theta_s$. The fair student is used to model the distribution of $P(\hat{y}|\mathbf{x})$ of the

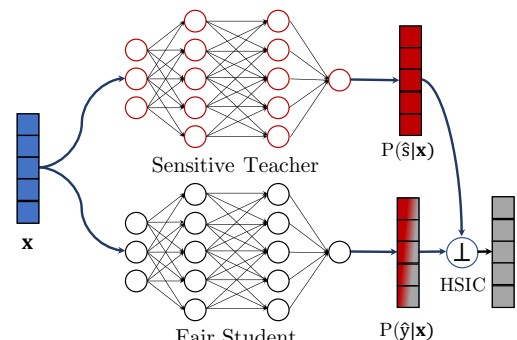

Figure 4: Overview of STFS. The sensitive teacher learns the distribution of sensitive information from non-sensitive attributes, and the fair student learns the distribution of the downstream task prediction. Then we encourage the independence between the outcome of sensitive teacher and fair student by enforcing the Hilbert-Schmidt Independence Criterion to be 0. The model will ensure $P_{,}(\hat{y}, s|x) = P(\hat{y}|x)P(s|x)$.

downstream task, which is $f_s(\mathbf{x}, \theta_s) = P(\hat{y}|\mathbf{x})$, where the trainable parameters are $\theta_s$.

**Training Procedure.** We pre-train the sensitive teacher $\hat{s} = f_t(\mathbf{x}, \theta_t)$ with the cross-entropy loss function $\mathcal{L}_{CE}(\hat{s}, s)$. Then, we infer the sensitive information $\hat{s}$ for all samples and use it to debias the fair student model. Next, we use the following objective function to optimize the fair student model:

$$
\begin{aligned}
Loss &= \mathcal{L}_{ce}(\hat{y}, y) + \lambda \cdot \mathcal{L}_{fair}(\hat{s}, \hat{y}) \\
&= \mathcal{L}_{ce}(\hat{y}, y) + \lambda \cdot \mathrm{HSIC}(\hat{s}, \hat{y}),
\end{aligned}
\tag{2}
$$

where $\hat{y} = f_s(\mathbf{x}, \theta_s) = P(y|\mathbf{x})$, $\hat{s} = f_t(\mathbf{x}, \theta_t) = P(s|\mathbf{x})$, and both of them are continuous values. The loss function $\mathcal{L}_{ce}$ optimizes the downstream task prediction, while $\mathrm{HSIC}(\cdot)$ encourages the distribution of sensitive information and prediction to be independent. The hyper-parameter $\lambda$ controls the balance between performance and fairness.

**Independence Guarantee.** We present an analysis of the independence guarantee of the distributions of sensitive information and prediction. In our method, we minimize the HSIC to ensure independence between the distributions of sensitive information and prediction. The Hilbert-Schmidt Independence Criterion (HSIC) is used to test if two random variables are independent, only with the data samples from the random variables. HSIC was introduced by Gretton et al. (2005b; 2008); Vepakomma et al. (2019). Consider two random variables $X$ and $Y$. HSIC (Gretton et al., 2005b) is defined as the Hilbert-Schmidt norm of the cross-covariance operator between the distributions $X$ and $Y$ in Reproducing Kernel Hilbert Space (RKHS):

$$
\begin{aligned}
\text{HSIC}(\mathbb{P}_{XY}, \mathcal{H}, \mathcal{G}) &= \|C_{XY}\|^2 \\
&= \mathbb{E}_{XYX'Y'}[k_X(X, X')k_{Y'}(Y, Y')] \\
&\quad + \mathbb{E}_{XX'}[k_X(X, X')]\mathbb{E}_{Y'}[k_Y(Y, Y')] \\
&\quad - 2\mathbb{E}_{XY}[\mathbb{E}_{X'}[k_X(X, X')]\mathbb{E}_{Y'}[k_Y(Y, Y')]],
\end{aligned}
\tag{3}
$$

where $k_X$ and $k_Y$ are kernel functions, $\mathcal{H}$ and $\mathcal{G}$ are the Hilbert spaces, and $\mathbb{E}_{XY}$ is the expectation over $X$ and $Y$. In practice, we can only observe the data samples while the exact distribution is unknown. Let $\mathcal{D} := \{(\mathbf{x}_1, \mathbf{y}_1), \cdots, (\mathbf{x}_m, \mathbf{y}_m)\}$ contain $m$ i.i.d. samples drawn from $\mathbb{P}_{XY}$, where $\mathbf{x}_i \in \mathbb{R}^{d_x}$ and $\mathbf{y}_i \in \mathbb{R}^{d_y}$. Then Equation (3) leads to the following empirical expression (Gretton et al., 2005a):

$$
\text{HSIC}(\mathcal{D}, \mathcal{H}, \mathcal{G}) = (m-1)^{-2}\, \text{tr}(\mathbf{K}_X \mathbf{H} \mathbf{K}_Y \mathbf{H})
\tag{4}
$$

where $\mathbf{K}_X \in \mathbb{R}^{m \times m}$ and $\mathbf{K}_Y \in \mathbb{R}^{m \times m}$ have entries $\mathbf{K}_{Xij} = k(\mathbf{x}_i, \mathbf{x}_j)$ and $\mathbf{K}_{Yij} = k(\mathbf{y}_i, \mathbf{y}_j)$, and $\mathbf{H} \in R^{m \times m}$ is the centering matrix $\mathbf{H} = \mathbf{I}_m - \frac{1}{m}\mathbf{1}_m\mathbf{1}_m^T$. With an appropriate kernel choice such as the Gaussian $k(\mathbf{x}, \mathbf{y}) \sim \exp(-\frac{1}{2}\|\mathbf{x} - \mathbf{y}\|^2/\sigma^2)$. HSIC is zero if and only if the random variables $X$ and $Y$ are independent, i.e., $P_{XY} = P_X P_Y$ (Sriperumbudur et al., 2010).

**Theorem 3.1.** *If STFS satisfies that HSIC($\hat{s}, \hat{y}$) = 0, then $\mathbb{E}_\alpha \alpha$-$\Delta DP$ = 0.*

*Proof Sketch:* The $\alpha$-sensitive group is decided by the distribution $\hat{s} = \text{P}(s|x)$. Thus sensitive information $\hat{y}_\alpha$ of $\alpha$-sensitive group is a function of $\hat{s}$. Since HSIC($\hat{s}, \hat{y}$) = 0, we have $\text{P}(\hat{y}, \hat{s}|x) = \text{P}(\hat{y}|x)\text{P}(\hat{s}|x)$. Combine these two, we have $\text{P}(\hat{y}, \hat{s}_\alpha|x) = \text{P}(\hat{y}|x)\text{P}(\hat{s}_\alpha|x)$, resulting in $\mathbb{E}_\alpha \alpha$-$\Delta DP$ = 0.

This theorem implies that our proposed method can effectively guarantee the achievement of $\alpha$-Demographic Parity, as it enforces the independence between the distribution of sensitive information and the model's predictions.

## 3.2 Discussion

In this section, we discuss the advantages of the proposed STFS method, as well as its potential limitations.

**Achieve Fairness with Limited Sensitive Attributes.** In real-world scenarios, achieving fairness with limited sensitive attributes is urgently needed, as sensitive attributes are typically very hard to collect, such as legal issues. Our proposed method is formulated as a cross-task knowledge distillation framework, the sensitive teacher can be trained with only a subset of the training samples. This feature makes our proposed method naturally applicable to limited sensitive attribute scenarios. We explore this more via an experiment with limited sensitive attributes in Section 4.4. The result shows that STFS can work well with a limited number of sensitive attributes.

**Relation to Demographic Parity.** Our proposed $\alpha$-Demographic Parity is closely related to the definition of demographic parity, thus we discuss the relation between our proposed fairness and demographic parity. Demographic Parity is a special case of $\alpha$-Demographic Parity. The reason is straightforward if we regard the whole sample as one group, the $\alpha$-Demographic Parity will be degraded to regular Demographic Parity. Our proposed $\alpha$-Demographic Parity is more strict fairness and has the following:

**Proposition 3.2.** *If a machine learning model satisfies the $\alpha$-Demographic Parity, it satisfies regular Demographic Parity.*

*Proof Sketch.* The binary sensitive attribute $s$ can be decided by the distribution $\text{P}(s|x)$, thus $s$ is a function of $\hat{s}$, i.e., $s = f(\hat{s})$. If a machine learning model satisfies $\alpha$-Demographic Parity, i.e., $\text{P}(\hat{y}, s|x) = \text{P}(\hat{y}|x)\text{P}(s|x)$.

**Limitations.** A possible limitation of our work is that model performance can be affected by the expressiveness of sensitive teacher models. Since the sensitive teacher is trained with the sensitive attributes as supervision, it could not be accurate to learn the distribution of sensitive information in non-sensitive attributes. The results show that our proposed method performs well in the case of limited sensitive attributes (sensitive teacher models may be considered undertrained).

## 4 Experiments

In this section, we evaluate the performance of the proposed method, STFS. First, we present the experimental setting, including the datasets, baselines, and implementation details. Then, we present the experimental results of the accuracy-fairness trade-off. We also present the experimental results on limited sensitive attribute scenarios. The major **observations** from the experiments are highlighted in boldface. We also evaluate our proposed method with the metric of demographic parity.

### 4.1 Experimental Setup

**Datasets.** Our experiment adopts the following datasets as our benchmark for evaluating the performance of our methods.

- **Adult** (Dua & Graff, 2017) contains personal information about $45,222$ individuals from the 1994 US Census. Each instance has 15 attributes, including race and gender. The downstream task of this dataset is to predict whether the income of an individual is greater than \$50k, which is shown to bias toward gender and race. We considered gender and race as sensitive attributes.

- **ACS-Income** (Ding et al., 2021) is from the American Community Survey (ACS) Public Use Microdata Sample (PUMS). Like UCI Adult, the downstream task of this dataset is to predict whether an individual's income is above \$50k too. The dataset contains $1,664,500$ data points. We also choose gender and race as sensitive attributes.

- **KDD Census** (Dua & Graff, 2017) contains $284,556$ instances with 41 attributes. The downstream task of this dataset is to predict whether the individual's income is above \$50k. The sensitive attributes for this dataset are gender and race.

**Baselines.** In our experiments, we use the following objective function $\mathcal{L}_{ce} + \lambda \mathcal{L}_{fair}$ to achieve fairness, where $\mathcal{L}_{ce}$ is the cross-entropy loss for the downstream task and $\mathcal{L}_{fair}$ is the constraint to ensure demographic parity. We adopt three regularizers as our baselines, and the details of them are as follows:

- **Gap** (Dua & Graff, 2017) is an in-process method that adds the violation of demographic parity regularization term to the objective function (Chuang & Mroueh, 2020; Kamishima et al., 2012). This method improves the fairness of the model with the regularization term simultaneously optimized during training, which takes $\Delta DP_c$ as the regularization term.

- **PRemover** (Prejudice Remover) (Kamishima et al., 2012) proposes prejudice remover as the regularization term, which enforces the independence between the prediction and sensitive attribute. PRemover minimizes mutual information to quantify the relation between the sensitive attribute and the prediction to ensure the independence between the prediction and the sensitive attribute.

- **HSIC** (Hilbert-Schmidt Independence Criterion) (Pérez-Suay et al., 2017; Quadrianto et al., 2019) uses HSIC as a regularization term to enforce the independence between model prediction and the sensitive attributes. Once the constraint HSIC equals 0, the mode prediction will be independent of explicit sensitive attributes [4].

**Implementation Details.** All the experiments are conducted on a server equipped with an NVIDIA RTX 3090 Ti GPU (24GB memory) and 256GB DDR4 memory. We use PyTorch (Paszke et al., 2019) to implement our code. Both the sensitive teacher and the fair student models are two-layer MLPs. We employ the Adam optimizer (Kingma & Ba, 2015) for training.

---

[4]The explicit sensitive attributes is different from sensitive information in non-sensitive attributes.

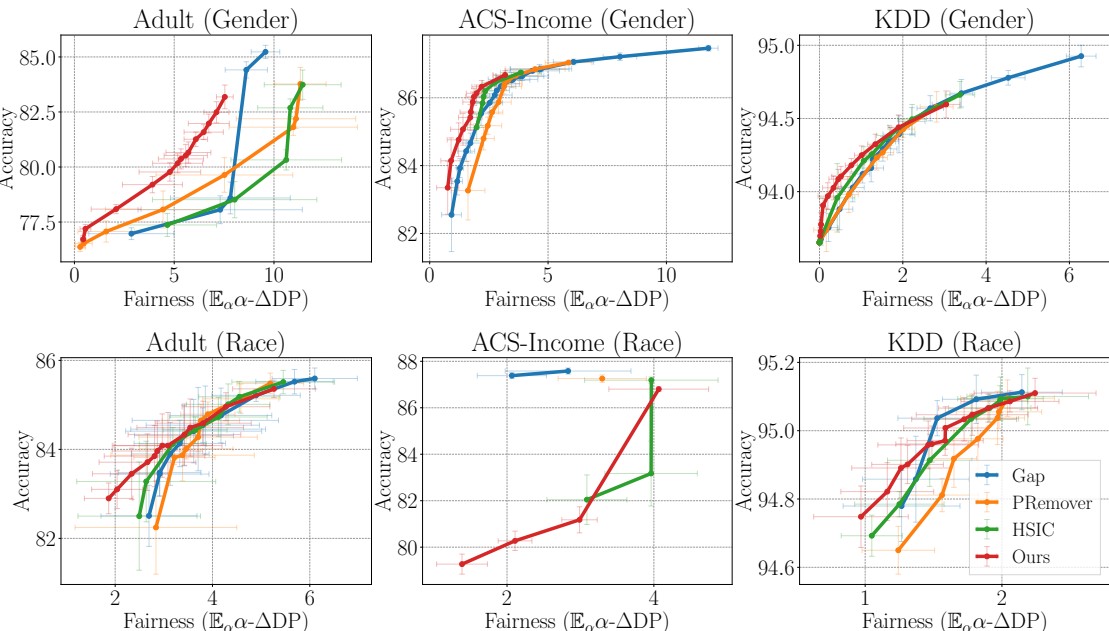

Figure 5: Pareto front for accuracy-fairness trade-off on multiple datasets with different sensitive attributes, i.e., gender and race. The x-axis represents the fairness metric $\mathbb{E}_\alpha \alpha$-$\Delta$DP, while the y-axis indicates the accuracy of the classification task. The ideal model should be situated in the top-left corner, simultaneously achieving the best prediction for downstream tasks and optimal fairness performance. Our results demonstrate that our method attains the most favorable accuracy-fairness trade-off. These figures are based on five runs with different seeds, and we report the mean and standard deviation of both $\mathbb{E}\alpha$-$\Delta$DP and accuracy.

**Evaluation.** For the performance of the downstream task, we use binary classification accuracy. To evaluate fairness performance, we utilize the proposed fairness metric, $\mathbb{E}_\alpha$-$\Delta$DP, which corresponds to $\alpha$-Demographic Parity. Furthermore, we also present the fairness performance in terms of $\Delta$DP.

## 4.2 Does STFS achieve $\alpha$-Demographic Parity?

We conducted experiments on various datasets to investigate the effectiveness of our proposed method in achieving $\alpha$-Demographic Parity, and the results are presented in Figure 5. We trained a set of models with different $\lambda$ values in Equation (2) and plotted the Pareto front for the accuracy-fairness trade-off. The results indicate that our method achieves a better Pareto front than other baselines, as the Pareto front of STFS is at the outermost edge in most cases. Therefore, we have **Observation 1: Our proposed STFS achieves the best trade-off between prediction accuracy and $\alpha$-Demographic Parity.**

## 4.3 How Does STFS Perform with Different $\alpha$?

In this experiment, we investigate the performance of our proposed method for different values of $\alpha$, and present the results in Figure 6. The baseline method used in this experiment employs Hilbert-Schmidt Independence Criterion (HSIC) as a regularization term to enforce independence between the model prediction and the binary sensitive attribute, which is the most similar method to our proposed method. The results demonstrate that our proposed method consistently achieves a lower $\alpha$-$\Delta$DP than the HSIC baseline, indicating that our method achieves better $\alpha$-Demographic Parity across various $\alpha$ values. Thus, we conclude that our proposed method outperforms the HSIC baseline and achieves a higher level of fairness. This leads to our **Observation 2: our proposed STFS achieves better $\alpha$-Demographic Parity across various $\alpha$ values.**

## 4.4 How Does STFS Perform with Limited Sensitive Attributes?

We perform experiments to demonstrate the effectiveness of our proposed method in scenarios with limited sensitive attributes. Concretely, we leverage partial training samples (ranging from 20% to 100%) to train

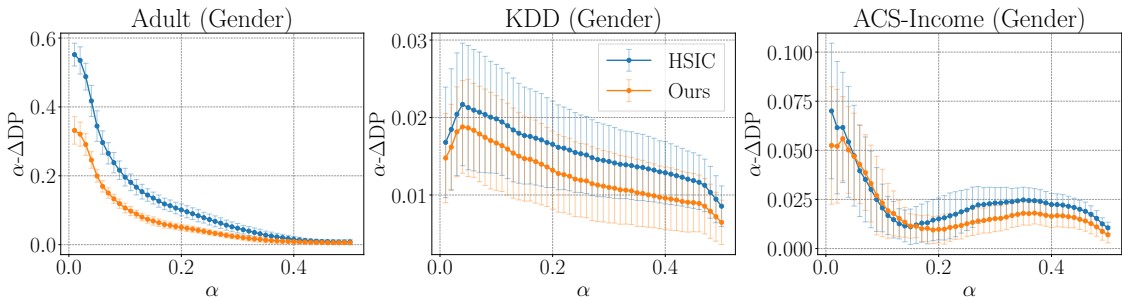

Figure 6: The performance of $\alpha$-Demographic Parity with varying $\alpha$ values. The sensitive attribute is "gender". A smaller $\alpha$ indicates a high level of sensitive information. The results show that our proposed method generally achieves lower $\alpha$-$\Delta$DP over varying $\alpha$ values. More results are presented in Appendix B.2.

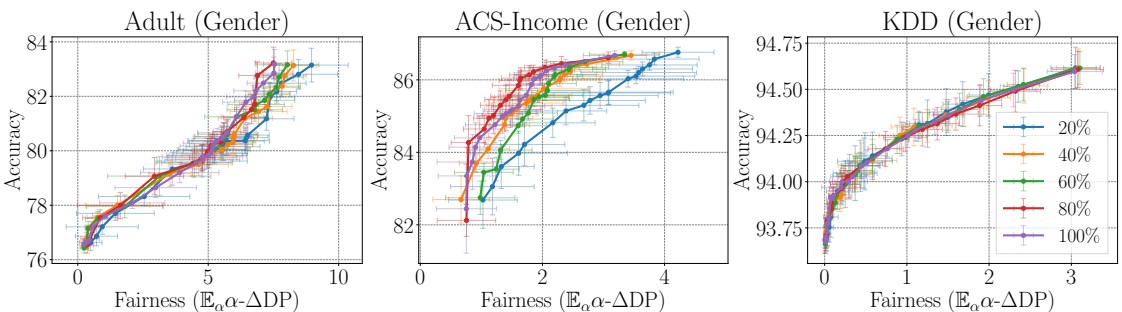

Figure 7: The Pareto front for accuracy-fairness trade-off on limited sensitive attributes scenarios. Models are trained with partial training samples, from 20% to 100%. The Pareto fronts are nearly the same line, demonstrating that sensitive teachers in STFS can be effectively trained by partial training samples. The sensitive attribute is gender. The figures are based on 5 runs with different seeds. The mean and standard deviation of $\mathbb{E}\alpha$-$\Delta$DP and accuracy are reported. More results are presented at Appendix B.3.

the sensitive teacher and use the sensitive teacher to predict the sensitive information for all training samples. Then we used predicted sensitive information to debias the fair student and reported the results in Figure 7. The results show that the Pareto fronts of the model trained with partial training samples (from 20% to 100%) are nearly the same line, demonstrating that sensitive teachers in STFS can be effectively trained by partial training samples. **Observation 3: Our proposed STFS can achieve $\alpha$-Demographic Parity with limited sensitive attributes.** This observation clearly demonstrates the superiority of our method on limited sensitive attributes scenarios.

## 5 How Does STFS Perform on Demographic Parity?

This section presents the performance evaluation of STFS, where we utilize *Demographic Parity* as the evaluation metric. We conducted two experiments to investigate the trade-off between accuracy and demographic parity, and the demographic parity performance in the limited sensitive attributes scenario.

### 5.1 The Performance of Accuracy-fairness Trade-off

We conducted experiments to evaluate the effectiveness of our proposed method in achieving a trade-off between prediction accuracy and demographic parity compared to several baseline methods. The results are presented in Figure 8. Our findings indicate that our method outperforms other baseline methods in achieving a better Pareto front in terms of demographic parity. Specifically, STFS is positioned at the outermost edge of the Pareto front in most cases. Therefore, we have that **Observation 4: our proposed STFS achieves the best trade-off between prediction accuracy and demographic parity among baseline methods.**

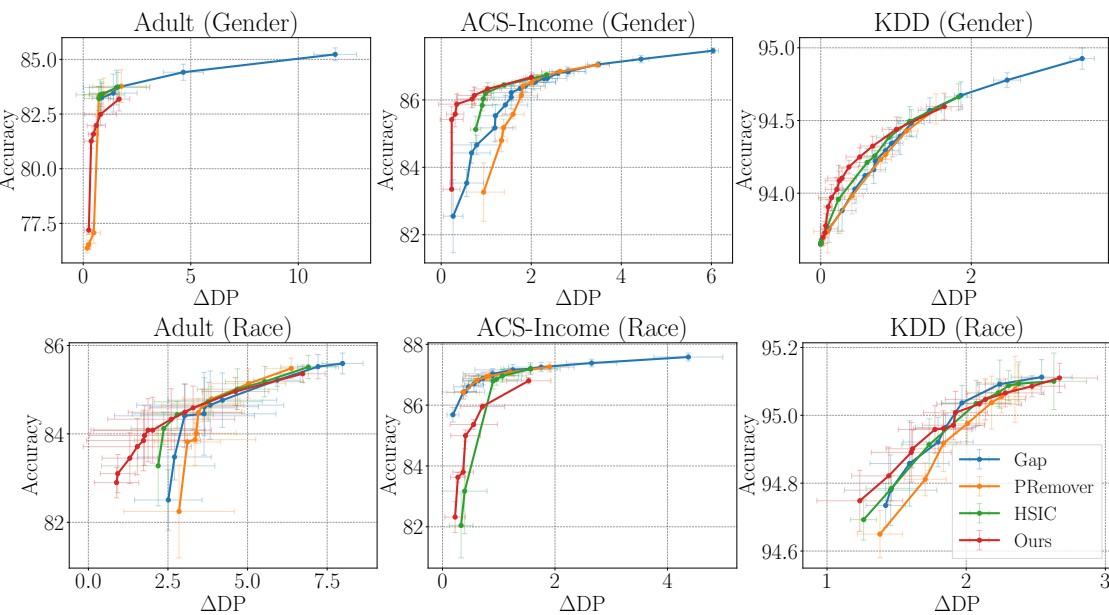

Figure 8: Pareto front for accuracy-fairness trade-off on multiple datasets with different sensitive attributes, i.e., gender and race. The fairness metric of the x-axis is the ΔDP. The accuracy of the y-axis is the performance of the classification task. The figures are based on 5 runs with different seeds. The mean and standard deviation of ΔDP and accuracy are reported. The results demonstrate that our method achieves a more favorable accuracy-fairness trade-off compared to baseline approaches in terms of ΔDP.

## 5.2 The Performance with Limited Sensitive Attribute

This section presents the experimental result to demonstrate the effectiveness of our proposed method in scenarios with limited sensitive attributes on the metric of demographic parity. We leverage partial training samples (ranging from 20% to 100%) to train the sensitive teacher and use the sensitive teacher to predict the sensitive information for all training samples. And we then used predicted sensitive information distribution for debiasing of the fair student model. The results are presented in Figure 9. The results show that the Pareto fronts of the model trained with partial training samples (from 20% to 100%) are nearly the same line, especially for the KDD dataset. We observed that **Observation 5: Our proposed method demonstrates superior performance in scenarios with limited sensitive attributes, achieving better demographic parity compared to alternative approaches.**

## 6 Related Work

In this section, we present the three lines of works related to our topic, including fairness, knowledge distillation, and Hilbert-Schmidt Independence Criterion.

**Fairness.** Recently, lots of works (Guldogan et al., 2023; Roh et al., 2023; Schrouff et al., 2022a;b; Vogel et al., 2020; Chouldechova & Roth, 2018; Roh et al., 2021; Zhang et al., 2022; Vogel et al., 2021; Maheshwari et al., 2022; Coston et al., 2020) from both academic and industrial have been proposed to address fairness issues. In this paper, we focus on in-processing fairness algorithms (Agarwal et al., 2018; Elkan, 2001; Jiang & Nachum, 2020; Kamishima et al., 2012; Zafar et al., 2017; Zhang et al., 2018), which leverage the fairness constraint to enforce the fairness when training a model. Among the fairness constraint method, statistical independence between the model's outputs and groups (Kamishima et al., 2012; Zafar et al., 2017) is a major approach. Besides, an adversarial learning technique is used to debias the model (Zhang et al., 2018), which makes the sensitive attribute unpredictable from the model by an adversary. In computer vision, the discrimination problem has usually been tackled in facial analysis, such as face recognition (Wang & Deng, 2020; Wang et al., 2019). Wang *et al.* (Wang et al., 2019) mitigated racial bias using domain adaptation technique and (Wang & Deng, 2020) utilized reinforcement learning.

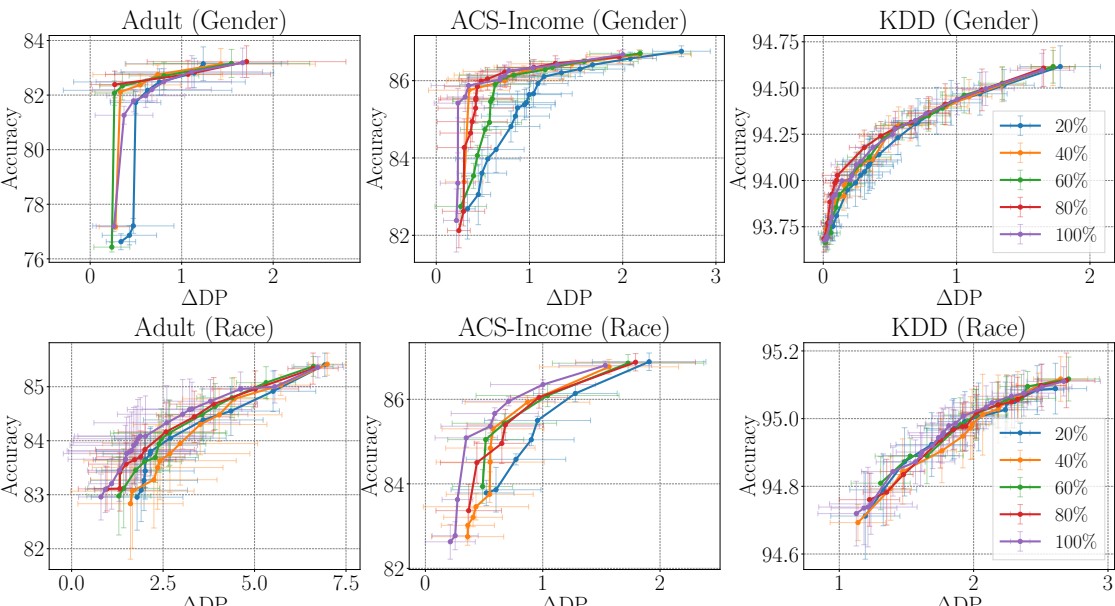

Figure 9: The Pareto front for accuracy-fairness trade-off in limited sensitive attributes scenarios. Models are trained with partial training samples, from 20% to 100%. The sensitive attribute is gender and race. The Pareto fronts are nearly the same line, demonstrating that sensitive teachers in STFS can be effectively trained by partial training samples in terms of $\Delta$DP.

**Knowledge Distillation.** Knowledge distillation facilitates the transfer of information from a teacher model to a student model. Following the pioneering work of (Hinton et al., 2015), in which the teacher model distills the softmax output distribution to the student, various extensions have focused on how to leverage the learned features. Romero *et al.* (Romero et al., 2015) introduced FitNet, which has the student mimic the teacher's features through linear regression. Zagoruyko *et al.* (Zagoruyko & Komodakis, 2017) proposed attention transfer (AT), which transfers knowledge using attention maps. Moreover, Yim et al. (2017) and Park et al. (2019) explored approaches utilizing the Gram matrix and relation map, respectively. Distinct from previous methods, Passalis & Tefas (2018) suggested techniques to minimize the distance between teacher and student feature distributions, as measured by the *Kullback-Leibler* divergence. Our proposed framework can be regarded as a cross-task knowledge distillation.

**HSIC.** The Hilbert-Schmidt Independence Criterion (HSIC) is a widely-used measure for independence and has been employed in robustness learning (Greenfeld & Shalit, 2020). Recently, HSIC has been applied to address fairness issues (Pérez-Suay et al., 2017; Quadrianto et al., 2019). For instance, Wu et al. (2018) explored the generalization properties of autoencoders using HSIC, while Lopez et al. (2018) utilized HSIC to restrict the latent space search and constrain the aggregate variational posterior. Furthermore, Vepakomma et al. (2019) utilized distance correlation, an alternative formulation of HSIC, to remove superfluous private information from medical training data. A notable advantage of HSIC is its ability to be computed using data samples rather than relying on the exact distribution of data. In our study, we employ HSIC to promote the independence between the distribution of sensitive information and prediction probabilities.

## 7  Conclusion

In this paper, we focus on investigating the violation of demographic parity. We observe from preliminary experiments that the different samples have different sensitive information leakage and diverse levels of violation of demographic parity. Based on this interesting observation, we propose from our different levels of alpha-demographic parity to measure the violation for specific sensitive information leakage groups. Additionally, we formulate a cross-task knowledge distillation framework to achieve $\alpha$-Demographic Parity via chasing the independence of the distribution of the sensitive information in non-sensitive attributes and that of downstream task prediction. Naturally, the proposed framework can also tackle the situation with limited sensitive attributes.

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

## A Proof of Theorem A.1

We first present the preliminary for the proof of Theorem A.1.

**Lemma A.1.** *If STFS satisfies that HSCI($\hat{s}, \hat{y}$) = 0, then $P(\hat{y}, s|\mathbf{x}) = P(\hat{y}|\mathbf{x})P(s|\mathbf{x})$*

*Proof:* Since we have HSIC is zero if and only if the random variables $X$ and $Y$ are independent, i.e., $P_{XY} = P_X P_Y$ (Sriperumbudur et al., 2010). The above theorem is easy to derive. This theorem suggests that if we minimize the HSIC regularization term to 0, the independence between the distributions of sensitive information and prediction will be guaranteed.

Next, we provide the proof of Theorem A.1. We begin with the following two statements:

1) Lemma 1 shows that if HSIC($\hat{s}, \hat{y}$) = 0, we have P($\hat{y}, \hat{s}|x$) = P($\hat{y}|x$)P($\hat{s}|x$).

2) The $\alpha$-sensitive group is decided by the distribution $\hat{s}$ = P($s|x$). Thus sensitive information $\hat{y}_\alpha$ of $\alpha$-sensitive group is a function of $\hat{s}$. The distribution of sensitive information for $\alpha$-sensitive group is a function of $\hat{s}$ = P($s|x$) since the $\alpha$-sensitive group is a subset of the entire samples. The function is the $|P(s|x) - 0.5| < \alpha$.

Combine the above two, we have P($\hat{y}, \hat{s}_\alpha|x$) = P($\hat{y}|x$)P($\hat{s}_\alpha|x$), resulting in $\mathbb{E}_\alpha \alpha\text{-}\Delta DP = 0$. ∎

## B Additional Experiments

In this appendix, we present the experimental results, which is the complementary experiment to the experimental results in the main body.

### B.1 Additional Experiments to Figure 2

In this subsection, we present the additional experimental results on another sensitive attribute, i.e., race, in Figure 10. The results also show that sensitive information in non-sensitive attributes varies across samples.

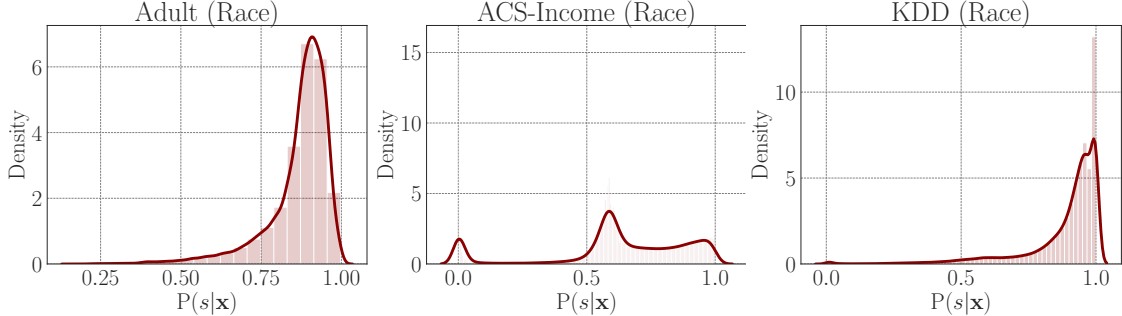

Figure 10: The distribution of P($s|\mathbf{x}$). The probability of P($s|\mathbf{x}$) is predicted by $\mathbf{x}$, which is the amount of *sensitive information*. The distribution shows that sensitive information in non-sensitive attributes varies over different samples. The sensitive attribute is race.

### B.2 Additional Experiments to Figure 6

In this experiment, we provide additional experiments to investigate the performance of our proposed model with different values of $\alpha$ and present the results in Figure 11. The results show that our proposed STFS generally obtains a lower $\alpha$-$\Delta$DP than baseline HSIC. The additional results show that our proposed STFS achieves better $\alpha$-Demographic Parity across various $\alpha$s.

### B.3 Additional Experiments to Figure 7

In this appendix, we present additional experimental results to demonstrate the effectiveness of our proposed method in scenarios with limited sensitive attributes. The experiments are conducted with race as the sensitive

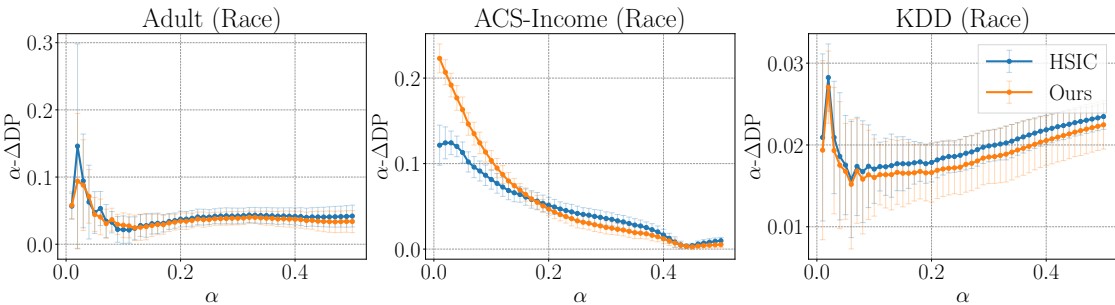

Figure 11: The performance of $\alpha$-Demographic Parity with different $\alpha$s. The sensitive attribute in the experiment is race. A smaller "alpha" indicates a high level of sensitive information. The results show that our proposed method generally achieves lower $\alpha$-$\Delta$DP over different $\alpha$s.

attribute. The results also show that the Pareto fronts of the models trained with varying percentages of training samples (from 20% to 100%) are nearly on the same line, demonstrating that our proposed STFS can achieve $\alpha$-Demographic Parity with limited sensitive attributes.

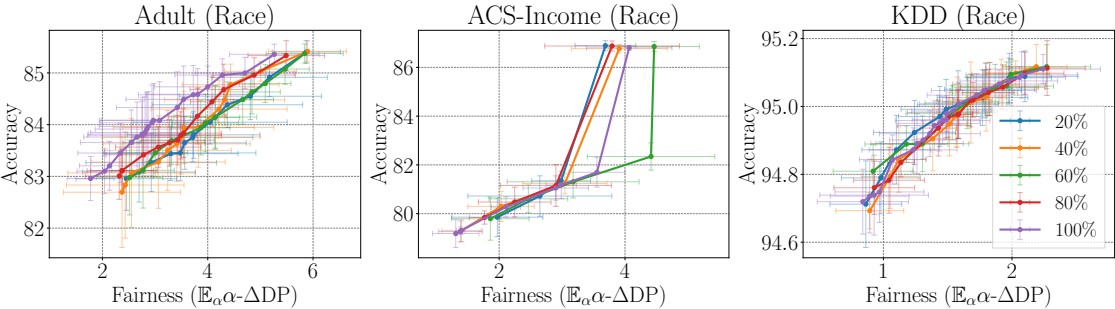

Figure 12: The Pareto front for the accuracy-fairness trade-off in scenarios with limited sensitive attributes. Models are trained using a varying percentage of training samples, ranging from 20% to 100%. The sensitive attribute is race.

