# OpenReview forum: "Do We Really Achieve Fairness with Explicit Sensitive Attributes?"
_TMLR — Withdrawn by Authors_

### Review · Reviewer_W34m · 2023-04-11

**Summary Of Contributions:**

This paper focuses on the fairness learning problem with the demographic parity metric. They argue that simply removing sensitive attributes from model inputs is not enough to achieve demographic parity, as non-sensitive attributes can still reveal sensitive information to varying extents. To this end, they introduce a new definition of group fairness measurement called α-Demographic Parity and propose to achieve it by directly promoting the independence of model predictions from the distribution of sensitive information.

The main contributions include:
1. A empirical study showing that non-sensitive attributes can reveal sensitive information to varying extents.
2. A new fairness measurement and a corresponding regularization term.
3. Experiment results showing the effectiveness of the proposed method.

**Audience:**

Yes

**Broader Impact Concerns:**

The research problem of this paper itself is about ethics, i.e., fairness. Thus, the limitation part should discuss the ethical implications rather than only the performance of the work. For example, while achieving a certain type of fairness, would it violate other types of fairness?

**Claims And Evidence:**

No

**Requested Changes:**

## The example 1 is not convincing
The authors intend to argue that **not all non-sensitive attributes can leak sensitive information to the same extent across all samples**. Then they consider an example of 'zipcode' being the non-sensitive attribute w.r.t. the event of acceptance in college admissions.

Instead of showing how the 'zipcode' leak different amounts of race information, they **directly stated** that `Since the “zipcode” leak different amounts of race information' and **suppose** one zipcode entirely leaks the sensitive attribute while another one leaks no race information.

It is a circular justification. I can't see how such a case supports your claim.

## The empirical study is problematic
The author trained a model to estimate the correlation between the sensitive attribute s (i.e., gender) and non-sensitive attributes x. Since the training method is not stated, I assume that they employ the same method as the training of the sensitive teacher by minimizing $L_{CE}(\hat{s}, s)$. In this way, $\hat{s}$ can be learned as the predicted probability of s given x. I can't see how such a $\hat{s}$ can measure the density distribution of P(s|x).

## The method is problematic
How do you choose the $\alpha$? What is the significance of achieving $\alpha-DP$? Let $\alpha=0.1$, then the 0.1-sensitive group contains high-sensitive samples, which could be samples with zipcode of African-American community. Then, the proposed can achieve fairness within this 0.1-sensitive group. However, only achieving fairness within this 0.1-sensitive group is meaningless. The predictor is still unfair to people with 'good' or 'bad' zipcode.

Besides, as mentioned above, the sensitive teacher is not properly learned.

## The experiment results are confusing
The α-∆DP by definition measures how far the **sensitive attributes** affect the **model prediction**. The baseline HSIC uses a regularization term to enforce the independence between **model prediction** and the **sensitive attributes** while the proposed one uses a regularization term to enforce the independence between **model prediction** and the **sensitive attributes distribution**. From my point of view, the baseline HSIC could have a smaller α-∆DP than the proposed method because the objective is exactly the measurement metric. Can the authors explain why the results are not like that?

## Typos
For example, the trainable parameters are $\theta_{t}$ for $f_t$; lacking punctuation at the end of Definition 2.1.

**Strengths And Weaknesses:**

Strengths:
1. The research problem is significant.
2. Examples and figures are included to help understand some ideas and concepts.

Weaknesses:
1. The writing, presentation, and organization can be improved. Lots of sentences are hard to read. Some sentences are not even completed, e.g., the bottom line on page 7 only contains an 'if'. Some technical terms are referred to without being defined, e.g., `ACM-Income' in Figure 1. I strongly suggest the authors revise the paper and make these sentences clear. In terms of the presentation and organization, I am concerned that the current version is insufficient to be published.
2. The empirical study and methodology are problematic. See requested changes for details.

---

### Review · Reviewer_qakX · 2023-04-14

**Summary Of Contributions:**

This paper studies how to enforce fairness when non-sensitive features exhibit the sensitive attribute leakage. For instance, in college admission, the zip code may reveal race information due to residential segregation and population distribution patterns. They propose $\alpha$ demographic parity, a new group fairness measure, by accounting for different levels of sensitive information leakage. To address the sensitive attribute leakage, they propose to train (1) a teacher model that can learn the sensitive information and (2) a student model to learn the targets. Then they use the Hilbert-Schmidt Independence Criterion to regularize the student model. Through extensive experiments, the authors show that

**Audience:**

Yes

**Broader Impact Concerns:**

I would suggest the authors to add a broader impact section to discuss any potential societal impacts of the studied $\alpha$-Demographic Parity measure as well as the STFS approach.

**Claims And Evidence:**

Yes

**Requested Changes:**

I would like to see the authors adequately addressed all the raised weaknesses above. The name of STFS is a bit weird, and I would like the authors to find a more interesting name.

**Strengths And Weaknesses:**

**Strengths**:
- Nice illustration of the sensitive attribute leakage problems through examples and figures.
- The experimental results are clearly discussed to analyze the performance of the proposed method.

**Weaknesses**:

The major weakness is that the proposed definitions of $\alpha$-Sensitive Group and $\alpha$-Demographic Parity, as described in Definition 2.1 and Definition 2.2, do not make sense to me.
- [Q1] Are the instances in the group with smaller $\alpha$ also inclusively in the group with larger $\alpha$? For example, if the instance has $P(s|x) = 0.1$, then the instance can belong to both 0.1-sensitive group and 0.2-sensitive group, since $0.1 \in [0, 0.2]$.
- [Q2] In Definition 2.2, $\alpha$ is not even a variable since you are saying *for all* $\alpha in [0, 0.5]$. Do you really mean for all $\alpha in [0, 0.5]$ or just any $\alpha \in [0., 0.5]$? And how is $\alpha$-$\Delta$DP related to $\alpha$-Demographic Parity?
- [Q3] In Equation (1), is $P(y | s=0)$ short for $P(y=1|s=0)$?

Other weaknesses:
The compared baseline methods are rather weak. There are many adversarial learning approaches very similar to the STFS approach. At least, the authors should compare with [1], who also proposed to train two networks (one for learning the labels, another for learning the sensitive attributes).

> [1] Brian Hu Zhang, Blake Lemoine, and Margaret Mitchell. 2018. Mitigating Unwanted Biases with Adversarial Learning. In Proceedings of the 2018 AAAI/ACM Conference on AI, Ethics, and Society (AIES '18). Association for Computing Machinery, New York, NY, USA, 335–340. https://doi.org/10.1145/3278721.3278779

---

### Review · Reviewer_sDXa · 2023-04-24

**Summary Of Contributions:**

This paper aims to achieve alpha-demographical parity for samples with differing degrees of sensitive information leakage by encouraging the independence of model predictions and the distribution of sensitive information. Experiments on the real datasets show some effectiveness of the proposed method. Additionally, the experiments on the scenarios with limited sensitive attributes also perform well.

**Audience:**

Yes

**Broader Impact Concerns:**

No concerns on the ethical implications of the work.

**Claims And Evidence:**

Yes

**Requested Changes:**

As discussed in the first weakness, the motivation is confused. It would be better to reorganize this paper.

**Strengths And Weaknesses:**

Strengths:

* The presentation is clear and easy to follow. The authors introduce the problem and then explain the proposed method in detail.

* The experiments are well-designed overall, and the results show some effectiveness of the proposed method.


Weaknesses:

* The motivation is confused. As far as I understand, demographic parity is a population level fairness criterion, which means it does not focus on removing bias for any of the individuals. According to the definition of demographic parity, employing explicit sensitive attributes is reasonable. Therefore, I don’t think achieving demographic parity by employing the explicit sensitive attributes has anything to do with differing degrees of sensitive information leakage. For me, alpha-demographic parity is like “subpopulation-level demographic parity”. As in Figure 8, when achieving demographic parity, there is no evidence showing HSIC using the explicit sensitive attributes is worse than the proposed method. In some of the cases, HSIC even performs better.

* Some of the experimental results are not significantly better than other baselines and there is no explanation for this in difference cases. For example, In Figure 5, for the dataset Adult, ACS-Income and KDD dataset with Race, the trade-off between accuracy and fairness Race of the proposed method is not that better and there is no explanation for this.

---

### Note · Authors · 2023-05-11

**Comment:**

We thank all the reviewers for their time and effort in reviewing our paper. We will carefully consider your valuable comments in our revision.

Sincerely,\
Authors

**Withdrawal Confirmation:**

I have read and agree with the venue's withdrawal policy on behalf of myself and my co-authors.